# Distributed Optimal Margin Distribution Machine

## Abstract

Optimal margin Distribution Machine (ODM), a newly proposed statistical learning framework rooting in the novel margin theory, demonstrates better generalization performance than the traditional large margin based counterparts. Nonetheless, the same with other kernel methods, it suffers from the ubiquitous scalability problem in terms of both computation time and memory. In this paper, we propose a Distributed solver for ODM (DiODM), which leads to nearly ten times speedup for training kernel ODM. It exploits a novel data partition method to make the local ODM trained on each partition has a solution close to the global one. When linear kernel used, we extend a communication efficient distributed SVRG method to further accelerate the training. Extensive empirical studies validate the superiority of our proposed method compared to other off-the-shelf distributed quadratic programming solvers for kernel methods.

## 1 Introduction

Recently, the study on margin theory (Gao & Zhou, 2013) demonstrates an upper bound disclosing that maximizing the minimum margin does not necessarily result in a good performance, and instead, the distribution rather than a single margin is much more important. Later on, the study on lower bound (Grønlund et al., 2019) further proves that the upper bound is almost optimal up to a logarithmic factor. Inspired by these insightful works, the Optimal margin Distribution Machine (ODM) is proposed (Zhang & Zhou, 2019), which explicitly optimizes the margin distribution by maximizing the mean and minimizing the variance simultaneously, and exhibits much better generalization performance than the traditional large margin based counterparts.Due to the superiority shown on both binary and multi-class classification tasks (Zhang & Zhou, 2014; 2017), many works attempt to extend ODM to more genreal learning settings, just to list a few, cost-sensitive learning (Zhou & Zhou, 2016; Cheng et al., 2017), weak supervised learning (Zhang & Zhou, 2018a;b; Luan et al., 2020; Zhang & Jin, 2020), multi-label learning (Tan et al., 2020), online learning (Zhang et al., 2020) and regression (Rastogi et al., 2020).

Plenty of successes on various learning tasks validate the superiority of this new statistical learning framework. However, these ODM based extensions suffer from the scalability problem because of both computation time and memory, the same as other kernel methods. Some works have devoted to accelerate the training of ODM for large scale data, e.g., CSVRG (Tan et al., 2019) takes advantage of the coreset method, whose main idea is to adaptively construct the landmark points to sketch the whole data. Other approximation methods being able to directly speed up ODM include random Fourier feature (Rahimi & Recht, 2007) and Nyström (Williams & Seeger, 2001). Notice that the random Fourier method adopts a data-independent kernel mapping while the Nyström method takes a distribution-unaware sampling, they are both inferior to coreset method due to the insufficient use of data, which may only lead to a degraded generalization performance. All these methods are proposed for computing on one cpu core, but with the dramatic progress of digital technologies, the data generated devices become as diverse as computers, mobile phones, smart watches, cars, etc, and the amount of data created each day grows tremendously, which one cpu core can hardly afford and motivates us to seek the distributed training methods.

Up to now many machine learning methods have already owned their distributed versions, among which the closest one to our problem is the distributed SVMs, whose ideas can be concluded into two classes. The first one works in distributed data level, i.e., dividing the data into partitions on

which local models are trained and then combined to produce the larger local or global models. For example, in (Graf et al., 2004; Hsieh et al., 2014; Singh et al., 2017), a tree architecture on partitions is designed first, guided by which the solutions of different partitions are aggregated; in (Yu et al., 2005; Navia-Vazquez et al., 2006; Loosli et al., 2007), some key instances identification and exchange are further introduced to accelerate the training; in (Si et al., 2017), both low-rank and clustering structure of the kernel matrix are considered to get an approximation of kernel matrix. The other one works in distributed optimization level, that is directly applying the distributed optimization method, such as augmented Lagrangian method (Forero et al., 2010) and alternating direction method of multipliers (Boyd et al., 2010), or extending existing solver to distributed environment, e.g. distributed SMO (Cao et al., 2006).

Although many distributed solver for quadratic programming (QP) problems can be directly applied to ODM, these off-the-shelf solvers all ignore the intrinsic structure of the problem and can hardly achieve the greatest efficiency. We propose a specially designed Distributed solver for ODM (DiODM) in this work. To be specific, we put forward a novel data partition method so that ODM trained on each partition has a solution close to that trained on the whole data. Therefore, when some partitions are merged to form a larger partition, the solution of ODM on it can be quickly obtained by concatenating the corresponding local solutions as the initial point. Besides, in the case of linear kernel being used, we extend a communication efficient distributed SVRG method to further accelerate the training. To summarize, the remarkable differences of DiODM compared with existing distributed QP solvers are threefold:

1) DiODM incorporates a novel partition strategy, which makes the solution of local ODM on each partition close to the global one so that the training can be accelerated.

2) When linear kernel is used, DiODM extends a communication efficient distributed SVRG method to further accelerate the training.

3) DiODM can maintain the generalization performance of ODM meanwhile achieve nearly ten times speedup, much more efficient than existing distributed QP solvers.

In the rest of this paper, some preliminaries are first introduced in section 2, followed by the technical detail of the proposed DiODM in section 3. In section 4, we present the results of experimental and give empirical observations. Last, in section 5 we conclude the paper with future work.

## 2 PRELIMINARIES

Before diving into the technical details, we first give some default notation in our paper. Sets are designated by upper case letters with mathcal font (e.g., $\mathcal{S}$). The input space is $\mathcal{X} \subseteq \mathbb{R}^N$ and $\mathcal{Y} = \{1, -1\}$ is output space. $\lceil \cdot \rceil$ is the round up function. For any positive integer $M$, the set of integers $\{1, \ldots, M\}$ is denoted by $[M]$. For the feature mapping $\phi : \mathcal{X} \mapsto \mathbb{H}$ associated to some positive definite kernel $\kappa$ where $\mathbb{H}$ is the corresponding reproducing kernel Hilbert space (RKHS), $\kappa(\boldsymbol{x}, \boldsymbol{z}) = \langle \phi(\boldsymbol{x}), \phi(\boldsymbol{z}) \rangle_{\mathbb{H}}$ holds for any $\boldsymbol{x}$ and $\boldsymbol{z}$. In this paper we consider the shift-invariant kernel such as RBF kernel for simplicity, which satisfies $\kappa(\boldsymbol{x}, \boldsymbol{z}) = \kappa(\boldsymbol{x} - \boldsymbol{z})$.

The traditional large margin based methods maximize the minimum margin, and consequently the obtained decision boundary is only determined by a fraction of instances with the minimum margin (Schölkopf & Smola, 2001), which may hurt the generalization performance.

On the other hand, ODM explicitly optimizes the margin distribution. Given a labeled data set $\{(\boldsymbol{x}_i, y_i)\}_{i \in [M]}$, ODM is initially formalized by maximizing the margin mean and minimizing the margin variance:

$$\min_{\boldsymbol{w}, \bar{\gamma}, \xi_i, \epsilon_i} \frac{1}{2} \|\boldsymbol{w}\|^2 - \eta \bar{\gamma} + \frac{\lambda}{2M} \sum_{i \in [M]} (\xi_i^2 + \epsilon_i^2) \quad \text{s.t.} \ \bar{\gamma} - \xi_i \le \gamma_i \le \bar{\gamma} + \epsilon_i, \ \forall i \in [M], \quad (1)$$

where $\eta$, $\lambda$ are the trading-off parameters, $\gamma_i = y_i \boldsymbol{w}^\top \phi(\boldsymbol{x}_i)$ is the margin of the labeled instance $(\boldsymbol{x}_i, y_i)$ and $\bar{\gamma}$ is the mean of $\gamma_i$. Note that the two slack variables $\xi_i$ and $\epsilon_i$ are deviations from the margin mean, therefore the third term is exactly the margin variance.

The margin mean can be fixed as 1 since scaling $\boldsymbol{w}$ does not affect the decision boundary. Besides, instances with margin less than 1 are more likely to be misclassified, therefore we introduce a new

hyper-parameter $\upsilon \in [0,1]$ to weight the $\xi_i$. Last but not least, notice that instances not lying on the hyperplane $y_i \boldsymbol{w}^\top \phi(\boldsymbol{x}_i) = 1$ all contribute to the decision boundary, to achieve a lightweight model, we can tolerate the deviation smaller than the given threshold $\theta$. Plugging all these back into Eqn. (1) yields the primal problem of ODM:

$$\min_{\boldsymbol{w},\xi_i,\epsilon_i} \ p(\boldsymbol{w}) = \frac{1}{2}\|\boldsymbol{w}\|^2 + \frac{\lambda}{2M}\sum_{i\in[M]}\frac{\xi_i^2 + \upsilon\epsilon_i^2}{(1-\theta)^2}$$

$$\text{s.t. } 1 - \theta - \xi_i \leq y_i\boldsymbol{w}^\top\phi(\boldsymbol{x}_i) \leq 1 + \theta + \epsilon_i, \ \forall i \in [M].$$

By introducing the Lagrange multipliers $\boldsymbol{\zeta}, \boldsymbol{\beta} \in \mathbb{R}_+^M$ for the $2M$ inequality constraints respectively, the dual problem of ODM is

$$\min_{\boldsymbol{\zeta},\boldsymbol{\beta}\in\mathbb{R}_+^M} d(\boldsymbol{\zeta},\boldsymbol{\beta}) = \frac{1}{2}(\boldsymbol{\zeta}-\boldsymbol{\beta})^\top\mathbf{Q}(\boldsymbol{\zeta}-\boldsymbol{\beta}) + \frac{Mc}{2}(\upsilon\|\boldsymbol{\zeta}\|^2 + \|\boldsymbol{\beta}\|^2) + (\theta-1)\mathbf{1}_M^\top\boldsymbol{\zeta} + (\theta+1)\mathbf{1}_M^\top\boldsymbol{\beta},$$

where $[\mathbf{Q}]_{ij} = y_iy_j\kappa(\boldsymbol{x}_i,\boldsymbol{x}_j)$ and $c = (1-\theta)^2/\lambda\upsilon$ is a constant. The detailed derivation can be found in supplementary. By denoting $\boldsymbol{\alpha} = [\boldsymbol{\zeta};\boldsymbol{\beta}]$, the dual ODM can be rewritten as a standard convex QP problem:

$$\min_{\boldsymbol{\alpha}\in\mathbb{R}_+^{2M}} \ f(\boldsymbol{\alpha}) = \frac{1}{2}\boldsymbol{\alpha}^\top\mathbf{H}\boldsymbol{\alpha} + \boldsymbol{b}^\top\boldsymbol{\alpha}, \tag{2}$$

in which

$$\mathbf{H} = \begin{bmatrix} \mathbf{Q} + Mc\upsilon\mathbf{I} & -\mathbf{Q} \\ -\mathbf{Q} & \mathbf{Q} + Mc\mathbf{I} \end{bmatrix}, \quad \boldsymbol{b} = \begin{bmatrix} (\theta-1)\mathbf{1}_M \\ (\theta+1)\mathbf{1}_M \end{bmatrix}.$$

Notice that Eqn. (2) only involves $2M$ decoupled box constraints, thus it can be efficiently solved by a dual coordinate descent method. To be specific, in each iteration, only one variable is selected to be updated while other variables are kept as constants, which yields the following univariate QP problem of $t$:

$$\min_t \ f(\boldsymbol{\alpha} + t\boldsymbol{e}_i) = \frac{1}{2}[\mathbf{H}]_{ii}t^2 + [\nabla f(\boldsymbol{\alpha})]_it + f(\boldsymbol{\alpha}), \tag{3}$$

which has a closed-form solution:

$$[\boldsymbol{\alpha}]_i \leftarrow \max([\boldsymbol{\alpha}]_i - [\nabla f(\boldsymbol{\alpha})]_i/[\mathbf{H}]_{ii}, 0).$$

## 3 PROPOSED METHOD

DiODM works in distributed data level, that is it works by dividing the data into partitions on which local models are trained and then used to find the larger local or global models. For simplicity, we assume there are $K = p^L$ partitions at first with the same cardinality $m$, i.e., $m = M/K$. The data set $\{(\boldsymbol{x}_i, y_i)\}_{i\in[M]}$ are ordered so that the first $m$ instances are on the first partition, and the second $m$ instances are on the second partition, etc. That is for any instance $(\boldsymbol{x}_i, y_i)$, the index of partition to which it belongs is $P(i) = \lceil (i-1)/m \rceil$.

Suppose $\{(\boldsymbol{x}_i^{(k)}, y_i^{(k)})\}_{i\in[m]}$ is the data on the $k$-th partition, then the local ODM on it is

$$\min_{\boldsymbol{\zeta}_k,\boldsymbol{\beta}_k\in\mathbb{R}_+^m} d_k(\boldsymbol{\zeta}_k,\boldsymbol{\beta}_k) = \frac{1}{2}(\boldsymbol{\zeta}_k - \boldsymbol{\beta}_k)^\top\mathbf{Q}^{(k)}(\boldsymbol{\zeta}_k - \boldsymbol{\beta}_k)$$

$$+ \frac{mc}{2}(\upsilon\|\boldsymbol{\zeta}_k\|^2 + \|\boldsymbol{\beta}_k\|^2) + (\theta-1)\mathbf{1}_m^\top\boldsymbol{\zeta}_k + (\theta+1)\mathbf{1}_m^\top\boldsymbol{\beta}_k,$$

where $[\mathbf{Q}^{(k)}]_{ij} = y_i^{(k)}y_j^{(k)}\kappa(\boldsymbol{x}_i^{(k)},\boldsymbol{x}_j^{(k)})$. This problem can be rewritten as a standard convex QP problem in the manner of Eqn. (2), and efficiently solved by dual coordinate descent method as Eqn. (3).

Once the parallel training of $p^L$ local ODMs are completed, we merge every $p$ partitions to form $p^{L-1}$ larger partitions. On each larger partition, a new local ODM is trained again by dual coordinate descent method, but the optimization procedure is not started from the scratch. Instead, the

$p$ solutions obtained on the $p$ corresponding smaller partitions making up this larger partition, are concatenated as the initial point of the optimization. Since by our proposed novel partition strategy, this concatenated solution is already a good approximation to the optimal solution of ODM on the larger partition. We will elaborate on this in subsequent sections.

The above procedure is repeated until the solution converges or all the partitions are merged together. We give the pseudo-code of DiODM in Algorithm 1.

---

**Algorithm 1** DiODM

---

**Input**: Data set $\mathcal{D} = \{(\boldsymbol{x}_i, y_i)\}_{i=1}^{M}$.
**Parameter**: Partition control parameter $p$, number of stratums $S$, number of iterations $L$.
**Output** The solution $\boldsymbol{\alpha}$.

 1: Get $S$ landmarks by Eqn. (5) on $\mathcal{D}$.
 2: Sample instances without replacement to get partitions $\mathcal{D}_1, \ldots, \mathcal{D}_{p^L}$.
 3: Initial the dual solution $\boldsymbol{\alpha}^L = \mathbf{0}$.
 4: **for** $l = L, \ldots, 1$ **do**
 5:    **if** $\alpha$ converge **then**
 6:       break
 7:    **end if**
 8:    **for** $p = 1, \ldots, p^l$ **do**
 9:       Solve the ODM with dual coordinate descent.
10:    **end for**
11:    Merge every $p$ partitions to form new partitions.
12:    $\boldsymbol{\alpha}^{l-1} = \boldsymbol{\alpha}^{l}$.
13: **end for**
14: **return** $\boldsymbol{\alpha}^l$.

---

### 3.1 CONVERGENCE

In this subsection, we present a theorem to guarantee the convergence of the proposed method. Notice that the optimization variables on each partition are decoupled, they can be jointly optimized by the following problem:

$$\min_{\boldsymbol{\zeta}, \boldsymbol{\beta} \in \mathbb{R}_+^M} \widetilde{d}(\boldsymbol{\zeta}, \boldsymbol{\beta}) = \frac{1}{2}(\boldsymbol{\zeta} - \boldsymbol{\beta})^\top \widetilde{\mathbf{Q}} (\boldsymbol{\zeta} - \boldsymbol{\beta}) + \frac{mc}{2}(\upsilon \|\boldsymbol{\zeta}\|^2 + \|\boldsymbol{\beta}\|^2) + (\theta - 1)\mathbf{1}_M^\top \boldsymbol{\zeta} + (\theta + 1)\mathbf{1}_M^\top \boldsymbol{\beta},$$

where $\widetilde{\mathbf{Q}} = \mathrm{diag}(\mathbf{Q}^{(1)}, \ldots, \mathbf{Q}^{(K)})$ is a block diagonal matrix. It can be seen that the smaller the $K$, the more close the above formula to ODM, and when $K = 1$, it exactly degenerates to ODM. Therefore, DiODM deals with ODM by solving a series of problems which approaches to it, and the solution of former problems can be helpful for the optimization of the latter problems.

**Theorem 1.** *Suppose the optimal solutions of ODM and its approximate problem, i.e., $\widetilde{d}(\boldsymbol{\zeta}, \boldsymbol{\beta})$, are $\boldsymbol{\alpha}^\star = [\boldsymbol{\zeta}^\star; \boldsymbol{\beta}^\star]$ and $\widetilde{\boldsymbol{\alpha}}^\star = [\widetilde{\boldsymbol{\zeta}}^\star; \widetilde{\boldsymbol{\beta}}^\star]$ respectively, then the gaps between these two optimal solutions satisfy*

$$0 \leq d(\widetilde{\boldsymbol{\zeta}}^\star, \widetilde{\boldsymbol{\beta}}^\star) - d(\boldsymbol{\zeta}^\star, \boldsymbol{\beta}^\star) \leq U^2 (Q + M(M - m)c),$$

$$\|\widetilde{\boldsymbol{\alpha}}^\star - \boldsymbol{\alpha}^\star\|^2 \leq \frac{U^2}{Mc\upsilon}(Q + M(M - m)c),$$

*where $Q = \sum_{i,j:P(i) \neq P(j)} |[\mathbf{Q}]_{ij}|$ is the sum of the absolute values of the $\mathbf{Q}$'s entries which turn to zero in $\widetilde{\mathbf{Q}}$, and $U = \max(\|\boldsymbol{\alpha}^\star\|_\infty, \|\widetilde{\boldsymbol{\alpha}}^\star\|_\infty)$ upperbounds the infinity norm of solutions.*

Due to the page limitations, we only provide the sketch of proof here. The full proof can be found in the supplementary.

*Proof sketch.* The left-hand side of the first inequality is due to the optimality of $\boldsymbol{\zeta}^\star$ and $\boldsymbol{\beta}^\star$.

By comparing the definition of $d(\boldsymbol{\zeta}, \boldsymbol{\beta})$ with that of $\widetilde{d}(\boldsymbol{\zeta}, \boldsymbol{\beta})$, we can easily find that the only differences are the change of $\mathbf{Q}$ to $\widetilde{\mathbf{Q}}$, and $M$ to $m$. Thus the gap between $d(\boldsymbol{\zeta}^\star, \boldsymbol{\beta}^\star)$ and $\widetilde{d}(\boldsymbol{\zeta}^\star, \boldsymbol{\beta}^\star)$ can

be upper bounded by the infinity norm of $\boldsymbol{\alpha}^\star$ and the sum of the absolute values of the $\mathbf{Q}$'s entries which turn to zero in $\widetilde{\mathbf{Q}}$. The gap between $d(\widetilde{\boldsymbol{\zeta}}^\star, \widetilde{\boldsymbol{\beta}}^\star)$ and $\widetilde{d}(\widetilde{\boldsymbol{\zeta}}^\star, \widetilde{\boldsymbol{\beta}}^\star)$ can be upper bounded in a similar way. Combining these together with the optimality of $\widetilde{\boldsymbol{\zeta}}^\star$ and $\widetilde{\boldsymbol{\beta}}^\star$, i.e., $\widetilde{d}(\widetilde{\boldsymbol{\zeta}}^\star, \widetilde{\boldsymbol{\beta}}^\star) \leq \widetilde{d}(\boldsymbol{\zeta}^\star, \boldsymbol{\beta}^\star)$, can yield the right-hand side of the first inequality.

Notice that $f(\widetilde{\boldsymbol{\alpha}}^\star)$ is a quadratic function, hence besides the gradient $\boldsymbol{g}$ and Hessian matrix $\mathbf{H}$, all its higher derivatives are all zero, and it can be precisely expanded at $\boldsymbol{\alpha}^\star$ as

$$f(\boldsymbol{\alpha}^\star) + \boldsymbol{g}^\top (\widetilde{\boldsymbol{\alpha}}^\star - \boldsymbol{\alpha}^\star) + (\widetilde{\boldsymbol{\alpha}}^\star - \boldsymbol{\alpha}^\star)^\top \mathbf{H} (\widetilde{\boldsymbol{\alpha}}^\star - \boldsymbol{\alpha}^\star),$$

in which $\boldsymbol{g}^\top (\widetilde{\boldsymbol{\alpha}}^\star - \boldsymbol{\alpha}^\star)$ is nonnegative according to the the first order optimality condition. Furthermore, $\mathbf{H}$ can be lower bounded by the sum of a positive semidefinite matrix and a scalar matrix:

$$\mathbf{H} \succeq \begin{bmatrix} \mathbf{Q} & -\mathbf{Q} \\ -\mathbf{Q} & \mathbf{Q} \end{bmatrix} + Mc\upsilon \begin{bmatrix} \mathbf{I} & \\ & \mathbf{I} \end{bmatrix}.$$

By putting all these together, we can show that $\|\widetilde{\boldsymbol{\alpha}}^\star - \boldsymbol{\alpha}^\star\|^2$ is upper bounded by $f(\widetilde{\boldsymbol{\alpha}}^\star) - f(\boldsymbol{\alpha}^\star)$, i.e., $d(\widetilde{\boldsymbol{\zeta}}^\star, \widetilde{\boldsymbol{\beta}}^\star) - d(\boldsymbol{\zeta}^\star, \boldsymbol{\beta}^\star)$, and with the right-hand side of the first inequality we can derive the second inequality. □

This theorem indicates that the gap between the optimal solutions of ODM and its approximation, that is the problem solved in each iteration of DiODM, depends on $M - m$ and $Q$. As the iteration going on, the partitions become larger and larger, then the number of instances $m$ on each partition tends to be the total number of instances $M$; Furthermore, the matrix $\widetilde{\mathbf{Q}}$ approaches to $\mathbf{Q}$ which makes $Q$ decrease. Therefore, the solution obtained in each iteration of DiODM is getting closer and closer to that of ODM, that is to say, our proposed algorithm converges.

## 3.2 PARTITION STRATEGY

In this section we detail the partition strategy, which plays an important role in our proposed method, since partition strategy can significantly affect the optimization efficiency. Up to now, most partition strategies utilize the clustering algorithms to form the partitions. For example, DC-SVM (Hsieh et al., 2014) adopts the kernel $k$-means and simply regards each cluster as a partition. However, ODM heavily depends on the mean and variance of training data. Directly treating clusters as partitions will lead to significant difference among the distribution of partitions and the whole data, which makes the local solutions on each partition are far from the global one.

To preserve the original distribution as much as possible, we borrow the idea from stratified sampling, i.e., we first divide the data set into some homogeneous strata, and then apply random sampling within each stratum. To be specific, suppose the goal is to generate $K$ partitions. We first choose $S$ landmark points $\{\phi(\boldsymbol{z}_s)\}_{s \in [S]}$ in RKHS, and then construct one stratum for each landmark point by assigning the rest instances to the stratum in which its nearest landmark point lies, i.e., the index of stratum $\boldsymbol{x}_i$ belongs to is

$$\varphi(i) = \arg\min_{s \in [S]} \|\phi(\boldsymbol{x}_i) - \phi(\boldsymbol{z}_s)\|.$$

For each stratum $\mathcal{C}_s$, we equally divide it into $K$ pieces by random sampling without replacement, and take one piece from each stratum to make up a partition and totally $K$ partitions are created.

The remaining question is how to select these landmark points. Obviously, they should be representative enough to sketch the whole data distribution. To this end, we introduce the minimal principal angle which is defined between different stratum:

$$\tau = \min_{i \neq j} \left\{ \arccos \frac{\langle \boldsymbol{x}, \boldsymbol{z} \rangle}{\|\boldsymbol{x}\|\|\boldsymbol{z}\|} \;\middle|\; \boldsymbol{x} \in \mathcal{C}_i, \boldsymbol{z} \in \mathcal{C}_j \right\}.$$

Apparently, the larger the angle, the higher variation among the strata, and the more representative each partition is, which is strictly described by the following theorem.

**Theorem 2.** *For shift-invariant kernel $\kappa$ with $\kappa(0) = r^2$, that is $\|\phi(\boldsymbol{x})\| = r$ for any $\boldsymbol{x}$. With the partition strategy described above, for any $k \in [K]$, we have*

$$d_k(\boldsymbol{\zeta}_k, \boldsymbol{\beta}_k) - d(\boldsymbol{\zeta}^\star, \boldsymbol{\beta}^\star) \leq \frac{U^2 M^2 c}{2} + 2UM + U^2 M^2 r^2 + U^2 r^2 \cos\tau(2\Theta - M^2),$$

*where $\Theta = \sum_{i,j \in [M], i \neq j} 1_{\varphi(i) \neq \varphi(j)}$, and $U$ is same with Theorem 1.*

*Proof sketch.* Follow the construction process mentioned in appendix, we generate a data set $\mathcal{D}'_k$ where $|\mathcal{D}'_k| = |\mathcal{D}| = M$. According to our settings, the number of instances in $|\mathcal{D}'_k|$ belong to the s-th stratum equals to $|\mathcal{C}_s|$. Give the definition of $d_k(\boldsymbol{\zeta}_k, \boldsymbol{\beta}_k)$ and $d'_k(\boldsymbol{\zeta}'_k, \boldsymbol{\beta}'_k)$ in appendix, we have $d'_k(\boldsymbol{\zeta}'_k, \boldsymbol{\beta}'_k) = d_k(\boldsymbol{\zeta}_k, \boldsymbol{\beta}_k)$ since $\mathcal{D}_k$ and $\mathcal{D}'_k$ share the same margin mean and variance under the same hyperplane.

Denote $\boldsymbol{\alpha}'_k = [\boldsymbol{\zeta}'_k; \boldsymbol{\beta}'_k]$, $\boldsymbol{\gamma}'_k = [\boldsymbol{\zeta}'_k - \boldsymbol{\beta}'_k]$. By contrasting the definition of $d'_k(\boldsymbol{\zeta}'_k, \boldsymbol{\beta}'_k)$ with that of $d(\boldsymbol{\zeta}^\star, \boldsymbol{\beta}^\star)$, we can find that the differences consists of three parts. The first part is

$$\frac{Mc}{2}(\upsilon\|\boldsymbol{\zeta}'_k\|^2 + \|\boldsymbol{\beta}'_k\|^2 - \upsilon\|\boldsymbol{\zeta}^\star\|^2 - \|\boldsymbol{\beta}^\star\|^2),$$

and the second part is

$$(\theta + 1)\mathbf{1}_M^\top(\boldsymbol{\beta}'_k - \boldsymbol{\beta}^\star) + (\theta - 1)\mathbf{1}_M^\top(\boldsymbol{\zeta}'_k - \boldsymbol{\zeta}^\star).$$

Note that both of above parts can be upper bounded by the infinity norm of $\boldsymbol{\alpha}'_k$. The third part can be denoted as

$$g_k(\boldsymbol{\gamma}'_k) = \frac{1}{2}\boldsymbol{\gamma}'_k{}^\top \mathbf{Q}'_k \boldsymbol{\gamma}'_k - \frac{1}{2}\boldsymbol{\gamma}^\star{}^\top \mathbf{Q}\boldsymbol{\gamma}^\star.$$

With the boundness that $-U^2 \leq \gamma_i^\star \gamma_j^\star, \gamma'_{ki}\gamma'_{kj} \leq U^2, 0 \leq \upsilon \leq 1, 0 \leq \theta \leq 1$, we have

$$g_k(\boldsymbol{\gamma}'_k) \leq U^2 \sum_{i,j\in[M]} (\kappa(\boldsymbol{x}'^{(k)}_i, \boldsymbol{x}'^{(k)}_j) - \kappa(\boldsymbol{x}_i, \boldsymbol{x}_j)) \tag{4}$$

Upper bound $\kappa(\boldsymbol{x}'^{(k)}_i, \boldsymbol{x}'^{(k)}_j) - \kappa(\boldsymbol{x}_i, \boldsymbol{x}_j)$ by minimal principal angle and the law of cosines, we can easily find that $g_k(\boldsymbol{\gamma}'_k) \leq U^2 M^2 r^2 + U^2 r^2 \cos\tau(2\Theta - M^2)$. By combining the upper bound of above three parts, we can complete the proof. □

As shown in theorem 2, we derive an upper bound on the gap between the optimal object value on $\mathcal{D}$ and on $\mathcal{D}_k$. Note that $2\Theta > M^2$ holds for any $s \in [S]$ when $|\mathcal{C}_s| < M/2$, which can be easily satisfied. Thus, we can get more approximate solution in each partition by maximizing the minimal principal angle $\tau$.

However, due to the high computation cost, it is impossible to directly maximize minimal principal angle. Instead, we implement a greedy iterative process:

$$\boldsymbol{z}_1 = \arg\max_{\boldsymbol{z}\in\mathcal{D}} \kappa(\boldsymbol{z}, \boldsymbol{z}), \ \boldsymbol{z}_s = \arg\max_{\boldsymbol{z}\in\mathcal{D}/\{\boldsymbol{z}_1,...,\boldsymbol{z}_{s-1}\}} SC_{s-1}(\boldsymbol{z}) \tag{5}$$

where $SC_s(\boldsymbol{z}) = \kappa(\boldsymbol{z}, \boldsymbol{z}) - \mathbf{K}_{\boldsymbol{z},s}^\top \mathbf{K}_s^{-1} \mathbf{K}_{\boldsymbol{z},s}$ is the Schur complement of a new candidate landmark $\boldsymbol{z}_{s+1}$ based on the landmarks $\{\boldsymbol{z}_1, ..., \boldsymbol{z}_s\}$ which have been already selected, $\mathbf{K}_{\boldsymbol{z},s} = [\kappa(\boldsymbol{z}, \boldsymbol{z}_1), ..., \kappa(\boldsymbol{z}, \boldsymbol{z}_s)]$, $\mathbf{K}_s \in \mathbb{R}^{s\times s}$ is defined as $[\mathbf{K}_s]_{i,j} = \kappa(\boldsymbol{z}_i, \boldsymbol{z}_j)$. In the process of solving $\boldsymbol{z}_s$, we actually seek the instance which maximize the determinant of kernel matrix construct by $\{\boldsymbol{z}_1, ..., \boldsymbol{z}_s\}$. It is obviously that the maximum determinant is achieved when $\phi(\boldsymbol{z}_s)$ is orthogonal to $\text{span}\{\phi(\boldsymbol{z}_1), ..., \phi(\boldsymbol{z}_{s-1})\}$. Therefore, we can maximize the minimal principal angle by choosing the instance with maximum Schur complement.

It is noteworthy that each partition generated by our proposed strategy extracts proportional instances from each stratum, thus preserves the distribution. Besides, compared with other partition strategies based on $k$-means (Singh et al., 2017), we consider not only in the original feature space, but also in the situation where data can hardly be linear separated. Last but not the least, our partition strategy has lower time complexity.

## 3.3 ACCELERATION FOR LINEAR KERNEL

The solving process of dual coordinate descent requires too much storage and computing resources, mainly caused by the enormous kernel matrix. It is noteworthy that when linear kernel is used, we can directly solve the primal form of ODM to avoid computing and storing kernel matrix. Denote the gradient of $p(\boldsymbol{w})$ on instance $(\boldsymbol{x}_i, y_i)$ as $\nabla p_i(\boldsymbol{w})$, we have

$$\nabla p_i(\boldsymbol{w}) = \boldsymbol{w} + \frac{\lambda(y_i\boldsymbol{w}^\top\boldsymbol{x}_i + \theta - 1)y_i\boldsymbol{x}_i 1_{i\in\mathcal{I}_1}}{(1-\theta)^2} + \frac{\lambda\upsilon(y_i\boldsymbol{w}^\top\boldsymbol{x}_i - \theta - 1)y_i\boldsymbol{x}_i 1_{i\in\mathcal{I}_2}}{(1-\theta)^2}$$

where $\mathcal{I}_1 = \{i \mid y_i \boldsymbol{w}^\top \boldsymbol{x}_i < 1 - \theta\}$, and $\mathcal{I}_2 = \{i \mid y_i \boldsymbol{w}^\top \boldsymbol{x}_i > 1 + \theta\}$.

Since the the objective function of ODM is differentiable, distributed SVRG (DSVRG) (Lee et al., 2017) can be exploited in this scenario. It generates a series of extra auxiliary data sets sampling from the the original data set without replacement which share the same data distribution as the whole data set, so that an unbiased estimation of the gradient can be computed. In each iteration, all nodes are joined together to compute the full gradient first. Then each node performs the iterative update of SVRG in serial in a "round robin" fashion, i.e., let all nodes stay idle except one node performing a certain steps of iterative updates using its local auxiliary data and passing the solution to the next node. We show the process of solving DiODM by DSVRG algorithm in Algorithm 2.

---

**Algorithm 2** Accelerated DiODM for linear kernel

---

**Input**: Training data set $\mathcal{D} = \{(\boldsymbol{x}_i, y_i)\}_{i=1}^M$
**Parameters**: Number of partitions $K$, number of stratums $S$, number of stages $S_t$, step size $\eta$, the number of iterations $T$.
**Output**: Solution at $S_t$ stage $\boldsymbol{w}^{S_t}$

1: Get $S$ landmark points by Eqn. (5) on $\mathcal{D}$.
2: Sample instances without replacement to get partitions $\mathcal{D}_1, \ldots, \mathcal{D}_K$
3: Generate the auxiliary array $\mathcal{R}_1, \ldots, \mathcal{R}_k$ where $\mathcal{R}_i = \{j | (\boldsymbol{x}_j, y_j) \in \mathcal{D}_i\}$
4: $s \leftarrow 1$.
5: **for** $l = 0, 1, \ldots, S_t - 1$ **do**
6:     The center node sends $\boldsymbol{w}^l$ to each node.
7:     **for** each node $j = 1, 2, \ldots, K$ in parallel **do**
8:         Compute $\boldsymbol{h}_j^l = \sum_{i \in \mathcal{D}_j} \nabla p_i(\boldsymbol{w}^l)$
9:         send $\boldsymbol{h}_j^l$ to center.
10:     **end for**
11:     $\boldsymbol{h}^l = \frac{1}{M} \sum_{j=1}^K \boldsymbol{h}_j^l$
12:     Initial the solutions $\boldsymbol{w}_0^{l+1} = \boldsymbol{w}^l$
13:     **for** $t = 0, 1, 2, \ldots, T - 1$ **do**
14:         Sample instances $(\boldsymbol{x}_i, y_i)$ from $\mathcal{D}_s$ where $i \in \mathcal{R}_s$
15:         $\boldsymbol{w}_{t+1}^{l+1} = \boldsymbol{w}_t^{l+1} - \eta \nabla p_i(\boldsymbol{w}_t^{l+1}) - \nabla p_i(\boldsymbol{w}^l) + \boldsymbol{h}^l)$
16:         $\mathcal{R}_s \leftarrow \mathcal{R}_s \backslash i$
17:         **if** $\mathcal{R}_s = \emptyset$ **then**
18:             send $\boldsymbol{w}_{t+1}^{l+1}$ to node $s + 1$.
19:             $s \leftarrow s + 1$.
20:         **end if**
21:     **end for**
22:     $\boldsymbol{w}^{l+1} = \boldsymbol{w}_T^{l+1}$
23: **end for**
24: **return** $\boldsymbol{w}^{S_t}$

---

## 4 EXPERIMENTS

In this section, we evaluate the performance of our proposed algorithms by comparing with other QP solvers.

### 4.1 SETUP

We evaluate the performance of our proposed algorithms on seven real-world data sets.[1] The statistics of these data sets are summarized in Table 1. All features are normalized into the interval $[0, 1]$. For each data set, eighty percent of instances are randomly selected as training data, while the rest are selected as testing data. All the experiments are performed on a Spark (Zaharia et al. (2012)) cluster with one master and five workers. Each machine is equipped with 16 Intel Xeon E5-2670 CPU cores and 64G RAM.

---

[1]https://www.csie.ntu.edu.tw/~cjlin/libsvmtools/datasets/

DiODM is compared with three state-of-the-art distributed QP solvers, that is, Cascade approach (Ca-ODM) (Graf et al. (2004)), DiP approach (DiP-ODM) (Singh et al. (2017)) and DC approach (DC-ODM) (Hsieh et al. (2014)). We denote our proposed dual coordinate descent based algorithm as DiODM and accelerating algorithm as acc-DiODM. Besides, in evaluating the efficiency of acc-DiODM, two state-of-the-art gradient based methods are implementd, that is SVRG method ($\text{ODM}_{svrg}$) (Johnson & Zhang (2013)) and CSVRG method ($\text{ODM}_{csvrg}$) (Tan et al. (2019)).

| Data sets | gisette | phishing | a7a | cod-rna | ijcnn1 | skin-nonskin | SUSY |
|---|---|---|---|---|---|---|---|
| #Instances | 7,000 | 11,055 | 32,561 | 59,535 | 141,691 | 245,057 | 5,000,000 |
| #Features | 5,000 | 68 | 123 | 8 | 22 | 3 | 18 |

Table 1: Data set statistics.

## 4.2 RESULTS WITH LINEAR KERNEL

Figure 1 presents the time cost and test accuracy on data sets with linear kernel. In the Figure 1(a)(e), we show the training speedup ratio with cores increasing from 1 to 32 for DiODM and acc-DiODM. When 32 cores used, DiODM achieves more than 9 times training speedup while acc-DiODM achieves over 5 times training speedup. In other six figures, we can see that both DiODM and acc-DiODM show highly competitive performance compared with other methods. Specifically, acc-DiODM achieves better test accuracy and faster training speed than Ca-ODM, DPP-ODM and DC-ODM. DiODM achieves the better test accuracy on 5 data sets and just slightly worse than DC-ODM on skin-nonskin data set. On time cost, DiODM achieves faster training speed on all 6 data sets.

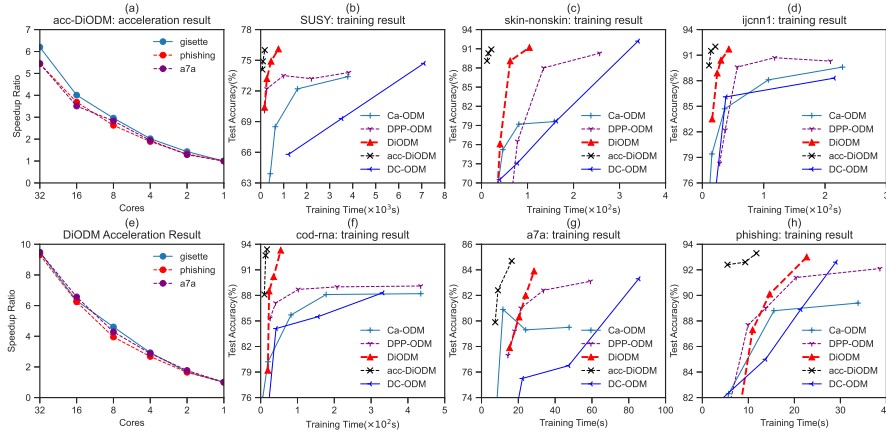

Figure 1: Comparisons of different methods using linear kernel. Each point of acc-DiODM except in Figure (a) indicates the result when every one third of stages executed. Other points except in Figure (e) indicate the result stop at different levels.

## 4.3 RESULTS WITH RBF KERNEL

Figure 2 presents the time cost and test accuracy on six data sets. It can be seen that DiODM shows highly competitive performance compared with other methods. Specifically, DiODM achieves the best test accuracy on 4 data sets and just slightly worse than DC-ODM on other 2 data sets. On time cost, DiODM achieves the fastest training speed on all 6 data sets.

## 4.4 COMPARISON WITH GRADIENT BASED METHODS

Figure 3 compares the training speed and generalization performance between our acceleration method and other gradient based methods. We observe that our method can get competitive test accuracy. Meanwhile, our method achieves over 5 times faster speed than other methods. This

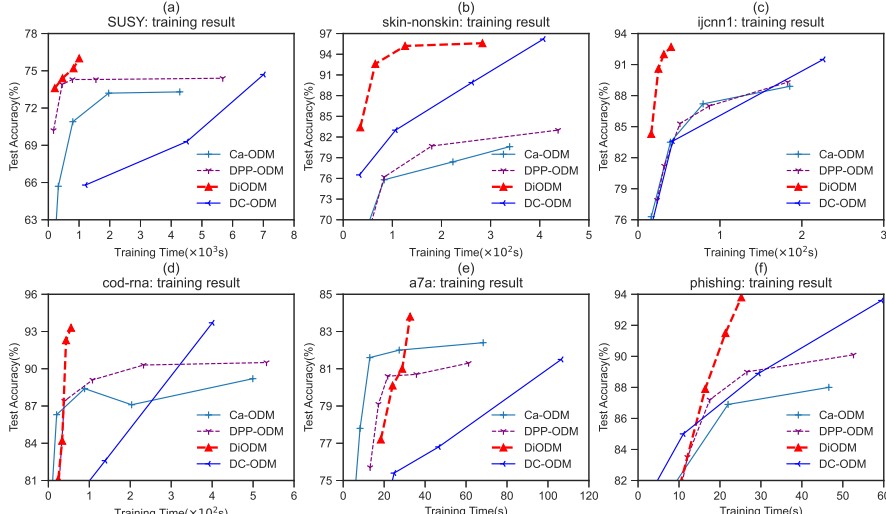

Figure 2: Comparisons of different methods using RBF kernel. Each point indicates the result when stop at different levels.

indicates that our distributed acceleration method achieves great training speed while hold the generalization performance.

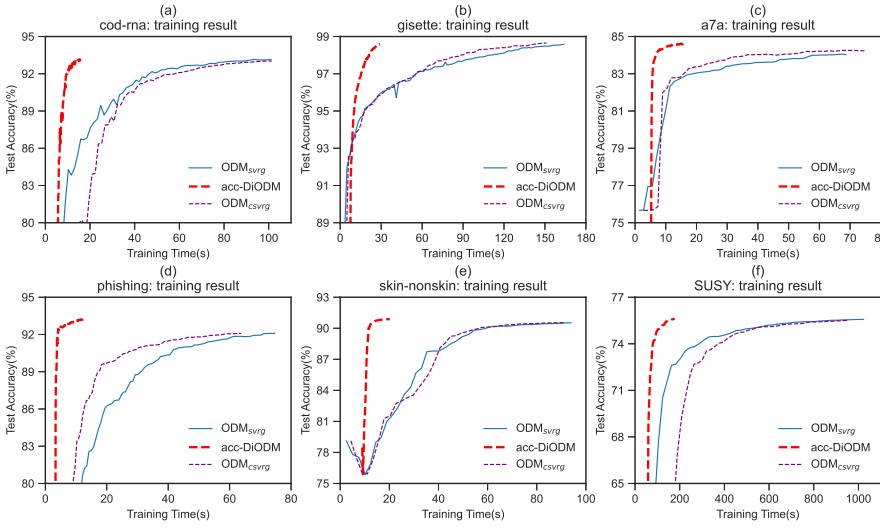

Figure 3: Comparisons of different gradient based methods.

## 5 CONCLUSION

While lots of works have been proposed to solve QP problems, these off-the-shelf solvers usually ignore the intrinsic structure of the problem, thus can hardly be implemented in ODM. We propose a distributed ODM solver called DiODM, in order to retain the first- and second- order statistics in both linear and nonlinear feature space. Additionally, an accelerating method is implemented to improve the training speed for linear kernel. According to our experiments, DiODM shows the superiority to other distributed QP solvers and generates great training acceleration in distributed environment. In the future, we will consider the circumstance in which data is located on different devices and can not be gathered together due to limited bandwidth or user privacy.

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

# A APPENDIX

## A.1 PRELIMINARIES

Given a labeled data set $\{(\boldsymbol{x}_1, y_1), \ldots, (\boldsymbol{x}_M, y_M)\}$, the primal problem of ODM is

$$\min_{\boldsymbol{w}, \xi_i, \epsilon_i} \ p(\boldsymbol{w}) = \frac{1}{2}\|\boldsymbol{w}\|^2 + \frac{\lambda}{2M} \sum_{i \in [M]} \frac{\xi_i^2 + v\epsilon_i^2}{(1-\theta)^2},$$

$$\text{s.t. } 1 - \theta - \xi_i \leq y_i \boldsymbol{w}^\top \phi(\boldsymbol{x}_i) \leq 1 + \theta + \epsilon_i, \ \forall i \in [M].$$

Denote $\mathbf{X} = [\phi(\boldsymbol{x}_1), \ldots, \phi(\boldsymbol{x}_M)]$, $\mathbf{Y} = \mathrm{diag}(y_1, \ldots, y_M)$, $\boldsymbol{\xi} = [\xi_1; \ldots; \xi_M]$, $\boldsymbol{\epsilon} = [\epsilon_1; \ldots; \epsilon_M]$, the above primal form can be rewritten as

$$\min_{\boldsymbol{w}, \boldsymbol{\xi}, \boldsymbol{\epsilon}} \ p(\boldsymbol{w}) = \frac{1}{2}\|\boldsymbol{w}\|^2 + \frac{\lambda(\|\boldsymbol{\xi}\|^2 + v\|\boldsymbol{\epsilon}\|^2)}{2M(1-\theta)^2}, \tag{6}$$

$$\text{s.t. } (1-\theta)\mathbf{1}_M - \boldsymbol{\xi} \leq \mathbf{Y}\mathbf{X}^\top \boldsymbol{w} \leq (1+\theta)\mathbf{1}_M + \boldsymbol{\epsilon},$$

where $\mathbf{1}_M$ is the $M$-dimensional all one vector.

With Lagrange multipliers $\boldsymbol{\zeta}, \boldsymbol{\beta} \in \mathbb{R}_+^M$ for the two constraints respectively, the Lagrangian of Eqn. (6) leads to

$$L = \frac{1}{2}\|\boldsymbol{w}\|^2 + \frac{\lambda(\|\boldsymbol{\xi}\|^2 + v\|\boldsymbol{\epsilon}\|^2)}{2M(1-\theta)^2} - \boldsymbol{\zeta}^\top (\mathbf{Y}\mathbf{X}^\top \boldsymbol{w} - (1-\theta)\mathbf{1}_M + \boldsymbol{\xi})$$

$$+ \boldsymbol{\beta}^\top (\mathbf{Y}\mathbf{X}^\top \boldsymbol{w} - (1+\theta)\mathbf{1}_M - \boldsymbol{\epsilon}), \tag{7}$$

and the KKT conditions are

$$\boldsymbol{w} = \mathbf{X}\mathbf{Y}(\boldsymbol{\zeta} - \boldsymbol{\beta}), \quad \boldsymbol{\xi} = \frac{M(1-\theta)^2}{\lambda}\boldsymbol{\zeta}, \quad \boldsymbol{\epsilon} = \frac{M(1-\theta)^2}{\lambda v}\boldsymbol{\beta}, \tag{8}$$

$$\zeta_i(y_i\boldsymbol{w}^\top\phi(\boldsymbol{x}_i) - (1-\theta) + \xi_i) = 0, \quad \beta_i(y_i\boldsymbol{w}^\top\phi(\boldsymbol{x}_i) - (1+\theta) - \epsilon_i) = 0. \tag{9}$$

Eqn. (8) is derived by setting the partial derivative of $L$ w.r.t. $\{\boldsymbol{w}, \boldsymbol{\xi}, \boldsymbol{\epsilon}\}$ to zero. Eqn. (9) is the complementary slackness conditions. Observe that $y_i\boldsymbol{w}^\top\phi(\boldsymbol{x}_i) < 1 - \theta$ and $y_i\boldsymbol{w}^\top\phi(\boldsymbol{x}_i) > 1 + \theta$ cannot hold simultaneously, therefore at least one of the two slack variables $\xi_i$ and $\epsilon_i$ is zero. According to Eqn. (8), we have $\zeta_i\beta_i = 0$ for any $i \in [M]$.

The following dual problem of ODM follows by substituting Eqn. (8) back into Eqn. (7):

$$\min_{\boldsymbol{\zeta}, \boldsymbol{\beta} \in \mathbb{R}_+^M} d(\boldsymbol{\zeta}, \boldsymbol{\beta}) = \frac{1}{2}(\boldsymbol{\zeta} - \boldsymbol{\beta})^\top \mathbf{Q}(\boldsymbol{\zeta} - \boldsymbol{\beta}) + \frac{Mc}{2}(v\|\boldsymbol{\zeta}\|^2 + \|\boldsymbol{\beta}\|^2) + (\theta - 1)\mathbf{1}_M^\top \boldsymbol{\zeta} + (\theta + 1)\mathbf{1}_M^\top \boldsymbol{\beta},$$

where $\mathbf{Q} = \mathbf{Y}\mathbf{X}^\top\mathbf{X}\mathbf{Y}$ and $c = (1-\theta)^2/\lambda v$ is a constant. By denoting $\boldsymbol{\alpha} = [\boldsymbol{\zeta}; \boldsymbol{\beta}]$, above problem can be rewritten as a standard convex quadratic programming:

$$\min_{\boldsymbol{\alpha} \in \mathbb{R}_+^{2M}} \ f(\boldsymbol{\alpha}) = \frac{1}{2}\boldsymbol{\alpha}^\top \begin{bmatrix} \mathbf{Q} + Mcv\mathbf{I} & -\mathbf{Q} \\ -\mathbf{Q} & \mathbf{Q} + Mc\mathbf{I} \end{bmatrix} \boldsymbol{\alpha} + \begin{bmatrix} (\theta - 1)\mathbf{1}_M \\ (\theta + 1)\mathbf{1}_M \end{bmatrix}^\top \boldsymbol{\alpha}.$$

Suppose the instances on the $k$-th partition are $\{(\boldsymbol{x}_1^{(k)}, y_1^{(k)}), \ldots, (\boldsymbol{x}_m^{(k)}, y_m^{(k)})\}$, then the dual problem of ODM on the $k$-th partition is

$$\min_{\boldsymbol{\zeta}_k, \boldsymbol{\beta}_k \in \mathbb{R}_+^m} d_k(\boldsymbol{\zeta}_k, \boldsymbol{\beta}_k) = \frac{1}{2}(\boldsymbol{\zeta}_k - \boldsymbol{\beta}_k)^\top \mathbf{Q}^{(k)}(\boldsymbol{\zeta}_k - \boldsymbol{\beta}_k)$$

$$+ \frac{mc}{2}(v\|\boldsymbol{\zeta}_k\|^2 + \|\boldsymbol{\beta}_k\|^2) + (\theta - 1)\mathbf{1}_m^\top \boldsymbol{\zeta}_k + (\theta + 1)\mathbf{1}_m^\top \boldsymbol{\beta}_k,$$

where $\mathbf{Q}_k = \mathbf{Y}_k\mathbf{X}_k^\top\mathbf{X}_k\mathbf{Y}_k$, $\mathbf{X}_k = [\phi(\boldsymbol{x}_1^{(k)}), \ldots, \phi(\boldsymbol{x}_m^{(k)})]$, and $\mathbf{Y}_k = \mathrm{diag}(y_1^{(k)}, \ldots, y_m^{(k)})$. Notice that the optimization variables $\boldsymbol{\zeta}_k$ and $\boldsymbol{\beta}_k$ are decoupled on each partition, by merging all the $K$ problems together, we can get the formulation of DiODM:

$$\min_{\boldsymbol{\zeta}, \boldsymbol{\beta} \in \mathbb{R}_+^M} \widetilde{d}(\boldsymbol{\zeta}, \boldsymbol{\beta}) = \frac{1}{2}(\boldsymbol{\zeta} - \boldsymbol{\beta})^\top \widetilde{\mathbf{Q}}(\boldsymbol{\zeta} - \boldsymbol{\beta}) + \frac{mc}{2}(v\|\boldsymbol{\zeta}\|^2 + \|\boldsymbol{\beta}\|^2) + (\theta - 1)\mathbf{1}_M^\top \boldsymbol{\zeta} + (\theta + 1)\mathbf{1}_M^\top \boldsymbol{\beta},$$

where $\widetilde{\mathbf{Q}} = \mathrm{diag}(\mathbf{Q}_1, \ldots, \mathbf{Q}_k)$ is a block diagonal matrix, $\boldsymbol{\zeta} = [\boldsymbol{\zeta}_1; \ldots; \boldsymbol{\zeta}_k]$, and $\boldsymbol{\beta} = [\boldsymbol{\beta}_1; \ldots; \boldsymbol{\beta}_k]$.

## A.2 PROOF OF THEOREM 1

**Theorem 1.** *Suppose the optimal solutions of ODM and DiODM are $\boldsymbol{\alpha}^\star = [\boldsymbol{\zeta}^\star; \boldsymbol{\beta}^\star]$ and $\widetilde{\boldsymbol{\alpha}}^\star = [\widetilde{\boldsymbol{\zeta}}^\star; \widetilde{\boldsymbol{\beta}}^\star]$, respectively. The gaps between the optimal objective values and solutions satisfy*

$$0 \leq d(\widetilde{\boldsymbol{\zeta}}^\star, \widetilde{\boldsymbol{\beta}}^\star) - d(\boldsymbol{\zeta}^\star, \boldsymbol{\beta}^\star) \leq U^2(Q + M(M - m)c), \tag{10}$$

$$\|\widetilde{\boldsymbol{\alpha}}^\star - \boldsymbol{\alpha}^\star\|^2 \leq \frac{U^2}{Mcv}(Q + M(M - m)c), \tag{11}$$

*where $U = \max(\|\boldsymbol{\zeta}^\star\|_\infty, \|\boldsymbol{\beta}^\star\|_\infty, \|\widetilde{\boldsymbol{\zeta}}^\star\|_\infty, \|\widetilde{\boldsymbol{\beta}}^\star\|_\infty)$, and $Q = \sum_{i,j:P(i)\neq P(j)} |[\mathbf{Q}]_{ij}|$.*

*Proof.* The left-hand side of Eqn. (10) is due to the optimality of $\boldsymbol{\zeta}^\star$ and $\boldsymbol{\beta}^\star$.

Without loss of generality, suppose the instances $\{(\boldsymbol{x}_1, y_1), \ldots, (\boldsymbol{x}_M, y_M)\}$ are ordered by partition index, i.e., the first $m$ instances are on the first partition, and the second $m$ instances are on the second partition, etc. According to the definition of $d(\boldsymbol{\zeta}, \boldsymbol{\beta})$ and $\widetilde{d}(\boldsymbol{\zeta}, \boldsymbol{\beta})$, and by denoting $\boldsymbol{\gamma} = \boldsymbol{\zeta} - \boldsymbol{\beta}$, we have

$$d(\boldsymbol{\zeta}, \boldsymbol{\beta}) = \widetilde{d}(\boldsymbol{\zeta}, \boldsymbol{\beta}) + \frac{1}{2}(\boldsymbol{\zeta} - \boldsymbol{\beta})^\top (\mathbf{Q} - \widetilde{\mathbf{Q}})(\boldsymbol{\zeta} - \boldsymbol{\beta}) + \frac{(M - m)c}{2}(v\|\boldsymbol{\zeta}\|^2 + \|\boldsymbol{\beta}\|^2)$$

$$= \widetilde{d}(\boldsymbol{\zeta}, \boldsymbol{\beta}) + \frac{1}{2}\sum_{i,j:P(i)\neq P(j)} \gamma_i \gamma_j [\mathbf{Q}]_{ij} + \frac{(M - m)c}{2}(v\|\boldsymbol{\zeta}\|^2 + \|\boldsymbol{\beta}\|^2).$$

In particular, the following holds:

$$d(\boldsymbol{\zeta}^\star, \boldsymbol{\beta}^\star) = \widetilde{d}(\boldsymbol{\zeta}^\star, \boldsymbol{\beta}^\star) + \frac{1}{2}\sum_{i,j:P(i)\neq P(j)} \gamma_i^\star \gamma_j^\star [\mathbf{Q}]_{ij} + \frac{(M - m)c}{2}(v\|\boldsymbol{\zeta}^\star\|^2 + \|\boldsymbol{\beta}^\star\|^2), \tag{12}$$

$$d(\widetilde{\boldsymbol{\zeta}}^\star, \widetilde{\boldsymbol{\beta}}^\star) = \widetilde{d}(\widetilde{\boldsymbol{\zeta}}^\star, \widetilde{\boldsymbol{\beta}}^\star) + \frac{1}{2}\sum_{i,j:P(i)\neq P(j)} \widetilde{\gamma}_i^\star \widetilde{\gamma}_j^\star [\mathbf{Q}]_{ij} + \frac{(M - m)c}{2}(v\|\widetilde{\boldsymbol{\zeta}}^\star\|^2 + \|\widetilde{\boldsymbol{\beta}}^\star\|^2). \tag{13}$$

Notice that at least one of $\zeta_i$ and $\beta_i$ is zero, thus $|\gamma_i| \leq |\zeta_i| + |\beta_i| \leq \max(|\zeta_i|, |\beta_i|) = U$. Subtracting Eqn. (12) from Eqn. (13) yields the right-hand side of Eqn. (10):

$$d(\widetilde{\boldsymbol{\zeta}}^\star, \widetilde{\boldsymbol{\beta}}^\star) - d(\boldsymbol{\zeta}^\star, \boldsymbol{\beta}^\star) \leq \frac{1}{2}\sum_{i,j:P(i)\neq P(j)} (\widetilde{\gamma}_i^\star \widetilde{\gamma}_j^\star - \gamma_i^\star \gamma_j^\star)[\mathbf{Q}]_{ij}$$

$$+ \frac{(M - m)c}{2}[v(\|\widetilde{\boldsymbol{\zeta}}^\star\|^2 - \|\boldsymbol{\zeta}^\star\|^2) + (\|\widetilde{\boldsymbol{\beta}}^\star\|^2 - \|\boldsymbol{\beta}^\star\|^2)]$$

$$\leq \frac{1}{2}\sum_{i,j:P(i)\neq P(j)} |\widetilde{\gamma}_i^\star \widetilde{\gamma}_j^\star - \gamma_i^\star \gamma_j^\star| \cdot |[\mathbf{Q}]_{ij}| + \frac{(M - m)c}{2}(v\|\widetilde{\boldsymbol{\zeta}}^\star\|^2 + \|\widetilde{\boldsymbol{\beta}}^\star\|^2)$$

$$\leq U^2 Q + U^2 M(M - m)c,$$

where the first inequality follows from the optimality of $\widetilde{\boldsymbol{\zeta}}^\star$ and $\widetilde{\boldsymbol{\beta}}^\star$, and the third inequality is derived by the boundness of $\boldsymbol{\zeta}, \boldsymbol{\beta}, \boldsymbol{\gamma}$ and $v \leq 1$.

Since $f(\boldsymbol{\alpha})$ is a quadratic function, it can be expanded at $\boldsymbol{\alpha}^\star$ as

$$f(\widetilde{\boldsymbol{\alpha}}^\star) = f(\boldsymbol{\alpha}^\star) + \nabla f(\boldsymbol{\alpha}^\star)^\top (\widetilde{\boldsymbol{\alpha}}^\star - \boldsymbol{\alpha}^\star) + (\widetilde{\boldsymbol{\alpha}}^\star - \boldsymbol{\alpha}^\star)^\top \nabla^2 f(\boldsymbol{\alpha}^\star)(\widetilde{\boldsymbol{\alpha}}^\star - \boldsymbol{\alpha}^\star)$$

$$\geq f(\boldsymbol{\alpha}^\star) + (\widetilde{\boldsymbol{\alpha}}^\star - \boldsymbol{\alpha}^\star)^\top \nabla^2 f(\boldsymbol{\alpha}^\star)(\widetilde{\boldsymbol{\alpha}}^\star - \boldsymbol{\alpha}^\star)$$

$$= f(\boldsymbol{\alpha}^\star) + (\widetilde{\boldsymbol{\alpha}}^\star - \boldsymbol{\alpha}^\star)^\top \begin{bmatrix} \mathbf{Q} + Mcv\mathbf{I} & -\mathbf{Q} \\ -\mathbf{Q} & \mathbf{Q} + Mc\mathbf{I} \end{bmatrix} (\widetilde{\boldsymbol{\alpha}}^\star - \boldsymbol{\alpha}^\star)$$

$$\geq f(\boldsymbol{\alpha}^\star) + (\widetilde{\boldsymbol{\alpha}}^\star - \boldsymbol{\alpha}^\star)^\top \begin{bmatrix} Mcv\mathbf{I} & \\ & Mc\mathbf{I} \end{bmatrix} (\widetilde{\boldsymbol{\alpha}}^\star - \boldsymbol{\alpha}^\star)$$

$$\geq f(\boldsymbol{\alpha}^\star) + Mcv\|\widetilde{\boldsymbol{\alpha}}^\star - \boldsymbol{\alpha}^\star\|^2,$$

where the first inequality follows from the first order optimality condition, and the third inequality uses the fact $v \leq 1$. Thus $\|\widetilde{\boldsymbol{\alpha}}^\star - \boldsymbol{\alpha}^\star\|^2$ can be upper bounded by

$$\|\widetilde{\boldsymbol{\alpha}}^\star - \boldsymbol{\alpha}^\star\|^2 \leq \frac{1}{Mcv}(d(\widetilde{\boldsymbol{\zeta}}^\star, \widetilde{\boldsymbol{\beta}}^\star) - d(\boldsymbol{\zeta}^\star, \boldsymbol{\beta}^\star)) \leq \frac{U^2}{Mcv}(Q + M(M - m)c),$$

which shows that Eqn. (11) holds and concludes the proof. $\qquad\square$

### A.3 PROOF OF THEOREM 2

**Theorem 2.** *For shift-invariant kernel $\kappa$ with $\kappa(0) = r^2$, that is $\|\phi(\boldsymbol{x})\| = r$ for any $\boldsymbol{x}$. With the partition strategy described above, for any $k \in [K]$, we have*

$$d_k(\boldsymbol{\zeta}_k, \boldsymbol{\beta}_k) - d(\boldsymbol{\zeta}^\star, \boldsymbol{\beta}^\star) \leq \frac{U^2 M^2 c}{2} + 2UM + U^2 M^2 r^2$$
$$+ U^2 r^2 \cos\tau(2\Theta - M^2)$$

*where $\Theta = \sum_{i,j \in [M], i \neq j} 1_{\varphi(\boldsymbol{x}_i) \neq \varphi(\boldsymbol{x}_j)}$*

*Proof.* Construct a data set $\mathcal{D}'_k$ based on partition $\mathcal{D}_k$ by repeating each instance $K$ times which appeared in $\mathcal{D}_k$, so that $|\mathcal{D}'_k| = |\mathcal{D}| = M$. According to our settings, the number of instances in $|\mathcal{D}'_k|$ belong to the s-th stratum equals to $|\mathcal{C}_s|$. Denote $\mathcal{D}'_k = \{(\boldsymbol{x}'^{(k)}_i, y'^{(k)}_i)\}_{i \in [M]}$ and guarantee that $\varphi(\boldsymbol{x}_i) = \varphi(\boldsymbol{x}'^{(k)}_i)$. Denote the dual form of ODM on $\mathcal{D}'_k$ as

$$\min_{\boldsymbol{\zeta}'_k, \boldsymbol{\beta}'_k \in \mathbb{R}^M_+} d'_k(\boldsymbol{\zeta}'_k, \boldsymbol{\beta}'_k) = \frac{1}{2}(\boldsymbol{\zeta}'_k - \boldsymbol{\beta}'_k)^\top \mathbf{Q}'_k(\boldsymbol{\zeta}'_k - \boldsymbol{\beta}'_k) + \frac{Mc}{2}(\upsilon\|\boldsymbol{\zeta}'_k\|^2 + \|\boldsymbol{\beta}'_k\|^2)$$
$$+ (\theta - 1)\mathbf{1}_M^\top \boldsymbol{\zeta}'_k + (\theta + 1)\mathbf{1}_M^\top \boldsymbol{\beta}'_k,$$

where $\mathbf{Q}'_k = \mathbf{Y}'_k \mathbf{X}'_k{}^\top \mathbf{X}'_k \mathbf{Y}'_k$, $\mathbf{X}'_k = [\phi(\boldsymbol{x}'_1{}^{(k)}), \ldots, \phi(\boldsymbol{x}'_m{}^{(k)})]$, and $\mathbf{Y}'_k = \mathrm{diag}(y'_1{}^{(k)}, \ldots, y'_m{}^{(k)})$. Denote the primal form of ODM on $\mathcal{D}'_k$ as

$$\min_{\boldsymbol{w}, \xi_i, \epsilon_i} p'_k(\boldsymbol{w}) = \frac{1}{2}\|\boldsymbol{w}\|^2 + \frac{\lambda}{2M}\sum_{i \in [M]} \frac{\xi'^2_i + \upsilon\epsilon'^2_i}{(1-\theta)^2},$$
$$\text{s.t. } 1 - \theta - \xi'_i \leq y'_i \boldsymbol{w}^\top \phi(\boldsymbol{x}'_i) \leq 1 + \theta + \epsilon'_i, \ \forall i \in [M].$$

For convenience, let $k = 1$ so that $\mathcal{D}_k = \{(\boldsymbol{x}_i, y_i)\}_{i \in [m]}$. From the primal problem of ODM on $\mathcal{D}_k$, we have

$$p_k(\boldsymbol{w}_k) = \frac{1}{2}\|\boldsymbol{w}_k\|^2 + \frac{\lambda}{2m}\sum_{i \in [m]} \frac{\xi^2_i + \upsilon\epsilon^2_i}{(1-\theta)^2} = \frac{1}{2}\|\boldsymbol{w}_k\|^2 + \frac{\lambda}{2M}\sum_{i \in [m]} K\frac{\xi^2_i + \upsilon\epsilon^2_i}{(1-\theta)^2}$$
$$= \frac{1}{2}\|\boldsymbol{w}_k\|^2 + \frac{\lambda}{2M}\sum_{i \in [M]} \frac{\xi'^2_i + \upsilon\epsilon'^2_i}{(1-\theta)^2} = p'_k(\boldsymbol{w}_k),$$

We can infer that the optimal solution of ODM on $\mathcal{D}_k$ and $\mathcal{D}'_k$ are the same. Therefore, $d'_k(\boldsymbol{\zeta}'_k, \boldsymbol{\beta}'_k) = d_k(\boldsymbol{\zeta}_k, \boldsymbol{\beta}_k)$. According to the definition,

$$d'_k(\boldsymbol{\zeta}'_k, \boldsymbol{\beta}'_k) - d(\boldsymbol{\zeta}^\star, \boldsymbol{\beta}^\star) = \frac{1}{2}(\boldsymbol{\zeta}'_k - \boldsymbol{\beta}'_k)^\top \mathbf{Q}'_k(\boldsymbol{\zeta}'_k - \boldsymbol{\beta}'_k) - \frac{1}{2}(\boldsymbol{\zeta}^\star - \boldsymbol{\beta}^\star)^\top \mathbf{Q}(\boldsymbol{\zeta}^\star - \boldsymbol{\beta}^\star)$$
$$+ \frac{Mc}{2}(\upsilon\|\boldsymbol{\zeta}'_k\|^2 + \|\boldsymbol{\beta}'_k\|^2 - \upsilon\|\boldsymbol{\zeta}^\star\|^2 - \|\boldsymbol{\beta}^\star\|^2)$$
$$+ (\theta - 1)\mathbf{1}_M^\top(\boldsymbol{\zeta}'_k - \boldsymbol{\zeta}^\star) + (\theta + 1)\mathbf{1}_M^\top(\boldsymbol{\beta}'_k - \boldsymbol{\beta}^\star)$$
$$= \frac{1}{2}\sum_{i,j \in [M]}(\gamma'_{ki}\gamma'_{kj}y'_i{}^{(k)}y'_j{}^{(k)}\kappa(\boldsymbol{x}'_i{}^{(k)}, \boldsymbol{x}'_j{}^{(k)}) - \gamma^\star_i\gamma^\star_j y_i y_j \kappa(\boldsymbol{x}_i, \boldsymbol{x}_j))$$
$$+ \frac{Mc\upsilon}{2}(\|\boldsymbol{\zeta}'_k\|^2 - \|\boldsymbol{\zeta}^\star\|^2) + \frac{Mc}{2}(\|\boldsymbol{\beta}'_k\|^2 - \|\boldsymbol{\beta}^\star\|^2)$$
$$+ (\theta - 1)\mathbf{1}_M^\top(\boldsymbol{\zeta}'_k - \boldsymbol{\zeta}^\star) + (\theta + 1)\mathbf{1}_M^\top(\boldsymbol{\beta}'_k - \boldsymbol{\beta}^\star)$$

$$(14)$$

and $-U^2 \leq \gamma^\star_i\gamma^\star_j, \gamma'_{ki}\gamma'_{kj} \leq U^2, 0 \leq \upsilon \leq 1, 0 \leq \theta \leq 1$. Then $d'_k(\boldsymbol{\zeta}'_k, \boldsymbol{\beta}'_k) - d(\boldsymbol{\zeta}^\star, \boldsymbol{\beta}^\star)$ can be upper bounded by

$$d'_k(\boldsymbol{\zeta}'_k, \boldsymbol{\beta}'_k) - d(\boldsymbol{\zeta}^\star, \boldsymbol{\beta}^\star) \leq U^2 \sum_{i,j \in [M]}(\kappa(\boldsymbol{x}'_i{}^{(k)}, \boldsymbol{x}'_j{}^{(k)}) - \kappa(\boldsymbol{x}_i, \boldsymbol{x}_j)) + \frac{U^2 M^2 c}{2} + 2UM \quad (15)$$

Notice that $\kappa(\boldsymbol{x}_i, \boldsymbol{x}_j) = \frac{1}{2}(\|\phi(\boldsymbol{x}_i)\|^2 + \|\phi(\boldsymbol{x}_j)\|^2 - \|\phi(\boldsymbol{x}_i) - \phi(\boldsymbol{x}_i)\|^2)$. Sequentially,

$$
\sum_{i,j \in [M]} (\kappa(\boldsymbol{x}'^{(k)}_i, \boldsymbol{x}'^{(k)}_j) - \kappa(\boldsymbol{x}_i, \boldsymbol{x}_j))
$$

$$
= \frac{1}{2} \sum_{i,j \in [M]} (\|\phi(\boldsymbol{x}_i) - \phi(\boldsymbol{x}_j)\|^2 - \|\phi(\boldsymbol{x}'^{(k)}_i) - \phi(\boldsymbol{x}'^{(k)}_j)\|^2)
$$

$$
= \frac{1}{2} \sum_{\varphi(\boldsymbol{x}_i) \neq \varphi(\boldsymbol{x}_j)} (\|\phi(\boldsymbol{x}_i) - \phi(\boldsymbol{x}_j)\|^2 - \|\phi(\boldsymbol{x}'^{(k)}_i) - \phi(\boldsymbol{x}'^{(k)}_j)\|^2)
$$

$$
+ \frac{1}{2} \sum_{\varphi(\boldsymbol{x}_i) = \varphi(\boldsymbol{x}_j)} (\|\phi(\boldsymbol{x}_i) - \phi(\boldsymbol{x}_j)\|^2 - \|\phi(\boldsymbol{x}'^{(k)}_i) - \phi(\boldsymbol{x}'^{(k)}_j)\|^2).
$$

For the situation where $\varphi(\boldsymbol{x}_i) \neq \varphi(\boldsymbol{x}_j)$, the maximal value of $\|\phi(\boldsymbol{x}_i) - \phi(\boldsymbol{x}_j)\|^2$ is $4r^2$, square of diameter of the ball. The minimal value of $\|\phi(\boldsymbol{x}_i) - \phi(\boldsymbol{x}_j)\|^2$ is $2r^2(1 - \cos\tau)$ which can be computed by the law of cosines since the angle between $\phi(\boldsymbol{x}_i)$ and $\phi(\boldsymbol{x}_j)$ is greater than $\tau$. According to the law of cosines, we can also upper bound $\|\phi(\boldsymbol{x}_i) - \phi(\boldsymbol{x}_j)\|^2$ by $2r^2(1 - \cos\tau)$ when $\varphi(\boldsymbol{x}_i) = \varphi(\boldsymbol{x}_j)$. It is obviously that $\|\phi(\boldsymbol{x}_i) - \phi(\boldsymbol{x}_j)\|^2 \geq 0$, thus

$$
\sum_{i,j \in [M]} (\kappa(\boldsymbol{x}'^{(k)}_i, \boldsymbol{x}'^{(k)}_j) - \kappa(\boldsymbol{x}_i, \boldsymbol{x}_j)) \leq \frac{1}{2} \sum_{\varphi(\boldsymbol{x}_i) \neq \varphi(\boldsymbol{x}_j)} (4r^2 - 2r^2(1 - \cos\tau))
$$

$$
+ \frac{1}{2} \sum_{\varphi(\boldsymbol{x}_i) = \varphi(\boldsymbol{x}_j)} 2r^2(1 - \cos\tau). \tag{16}
$$

Substitute Eqn. (16) into Eqn. (15), we can derive the bound

$$
d'_k(\boldsymbol{\zeta}'_k, \boldsymbol{\beta}'_k) - d(\boldsymbol{\zeta}^\star, \boldsymbol{\beta}^\star) \leq U^2 \sum_{\varphi(\boldsymbol{x}_i) \neq \varphi(\boldsymbol{x}_j)} ((2r^2 - r^2(1 - \cos\tau)) + \frac{U^2 M^2 c}{2}
$$

$$
+ \sum_{\varphi(\boldsymbol{x}_i) = \varphi(\boldsymbol{x}_j)} r^2(1 - \cos\tau)) + 2UM
$$

$$
\leq U^2 r^2 \Theta(1 + \cos\tau) + U^2 r^2 (M^2 - \Theta)(1 - \cos\tau) + \frac{U^2 M^2 c}{2} + 2UM
$$

$$
= U^2 M^2 r^2 + U^2 r^2 \cos\tau(2\Theta - M^2) + \frac{U^2 M^2 c}{2} + 2UM
$$

$\square$

