# OpenReview forum: "Distributed Optimal Margin Distribution Machine"
_ICLR.cc/2022/Conference — ICLR 2022 Submitted_

### Official Review · Reviewer_NozY · 2021-10-30

**Correctness:** 4
**Technical Novelty And Significance:** 3
**Empirical Novelty And Significance:** 3
**Recommendation:** 8
**Confidence:** 4

**Main Review:**

Pros
1.	The paper proposes a novel partition strategy for training kernel method in distributed scenario, leading to nearly ten times speedup.
2.	The paper is thorough and accomplished. Both linear and nonlinear kernels are fully considered. Both theoretical analyses and empirical studies are clearly presented.
3.	The experiments are conducted on data sets obtained from public repositories and necessary links to the readers are given.

Questions during rebuttal period:
Many studies have been devoted to reduce the time cost of kernel method, e.g., random feature, Nystrom method, etc. What is the advantage of DiODM compared to these off-the-shelf kernel acceleration methods?


**Summary Of The Paper:**

This paper proposes a distributed optimal margin distribution machine (DiODM), which can significantly reduce the training time so as to handle large scale data. Specifically, when nonlinear kernel applied, DiODM can lead to nearly ten times speedup based on the proposed novel partition strategy; as for linear kernel, DiODM extends a communication efficient distributed SVRG to further accelerate the training. The paper also presents the theoretical analysis inspiring the partition strategy, and extensive experimental results verifying the superiority of the proposed algorithm.

**Summary Of The Review:**

Generally, this is a solid work for accelerating ODM. The authors not only propose a novel partition strategy leading to excellent performance, but also detail the intuition behind, i.e., utilizing the stratified sampling to preserve data distribution. The theoretical analyses are broken down into many small steps and easy to follow.
The experiments are conducted on data sets obtained from public repositories and necessary links to the readers are given. Considering all these, I vote for acceptance.

---

> ### Author Response · Authors · 2021-11-15
> **Response to reviewer NozY**
>
> We thank the reviewer for the comments. We have revised the paper according to the suggestions and would like to clarify several things:
>
> [Q] Many studies have been devoted to reduce the time cost of kernel method, e.g., random feature, Nystrom method, etc. What is the advantage of DiODM compared to these off-the-shelf kernel acceleration methods?
>
> [A] The two methods mentioned by reviewer cannot be directly compared to our DiODM. Actually, among them is not competitive but complementary relationship, that is, they can be applied on each partition in our method to further accelerate the training.

---

### Official Review · Reviewer_jk73 · 2021-10-30

**Correctness:** 3
**Technical Novelty And Significance:** 3
**Empirical Novelty And Significance:** 3
**Recommendation:** 8
**Confidence:** 4

**Main Review:**


Strengths:
Overall, the paper proposes a solid contribution to a problem with broad applications, e.g., classification. They also show the effectiveness of their methods both theoretically and emprically. Moreover, the paper is pretty well-written and easy to follow. It is worth noting that tey authors make the presentation of equations and theoretical material accessible by providing the optimal levels of details so that they convey the main points without being tedious.

Weaknesses:

Despite many strengths, there are a few issues with the paper as I explain below:

(a) In page 2, regarding the sentence "therefore the third term is exactly the margin variance." it is my understanding the third term is the upper bound on the margin variance, not the margin variance itself.

(b) It is my understanding the Theorem 1 only implies that the difference between optimal solutions and optimal values of the main QP problem and partitioned QP problem are upper-bounded. What does this Theorem imply about the optimality of the solution obtained by running Alg. 1 with L iterations and S stratums?

(c) It seems that as the iterations grow the local QP problems on each partition get bigger, which I believe is a weakness of the proposed method. In fact, related to (b), it would be beneficial to know that after how many iterations we could stop the algorithm, while the local problems are not prohibitively large.

(d) What are the parameters (e.g., L and S in Alg. 1 and T, S, and K in Alg. 2) used in the experiments?










**Summary Of The Paper:**

The paper focuses on the Optimal margin Distribution Machine (ODM), i.e., a more robust version of the traditional maximum margin problems such as training Support Vector Machines (SVM). In particular, the authors introduce a novel iterative distributed algorithm for solving the resulting Quadrating Programming (QP) problem. They furthermore propose a method for optimally partitioning the data. In the case of linear kernels, they show that their method can benefit from the SVRG gradient methods. The proposed methods come with theoretical guarantees and they show that the proposed methods outperform or have competitive performance with three different QP solvers on seven real datasets.

**Summary Of The Review:**

Overall, I believe the authors address a generic and important problem. The proposed methods are novel and are supported with acceptable theoretical and empirical evidence. I believe the issues I raised above are only minor, but important to address, if possible.

---

> ### Author Response · Authors · 2021-11-15
> **Response to reviewer jk73**
>
> We thank the reviewer for the comments. We have revised the paper according to the suggestions and would like to clarify several things:
>
> [Q]  It is my understanding the third term is the upper bound on the margin variance, not the margin variance itself.
>
> [A] The third term is exactly the margin variance according to the original paper of ODM.
>
> [Q] What does this Theorem 1 imply about the optimality of the solution obtained by running Alg. 1 with L iterations and S stratums?
>
> [A] Your understanding is correct. In Alg. 1, $m = M / p^l$. Thus, when $l$ decreases, the solution in each partition gets closer to global solution.
>
> [Q] In fact, related to (b), it would be beneficial to know that after how many iterations we could stop the algorithm, while the local problems are not prohibitively large.
>
> [A] We describe an early stop mechanism in line 5-7 of Alg.1. In our implementation, this mechanism is performed when the change of the solution is less than a given threshold during the dual coordinate descent process after partition combined.
>
> [Q] What are the parameters (e.g., L and S in Alg. 1 and T, S, and K in Alg. 2) used in the experiments?
>
> [A] In Alg. 1 and Alg. 2, we select S from the set of $\{2^5, 2^6, …, 2^{10}\}$. In Alg. 1, L is select as $\{2^5\}$. In Alg. 2, parameter $K$ is related with our clusters. For dataset ijcnn1, skin-nonskin and SUSY, we set $K$ as 1000. For other datasets, we set $K$ as 200. We follow the set of parameter $T$ in DSVRG where $T*K = M$.

---

> > ### Comment · Reviewer_jk73 · 2021-11-28
> > **Thank You for the Response!**
> >
> > I want to thank the authors for their response and confirm that I have read their response. The authors partially answer my questions; what is missing is some theoretical guarantees on the solution obtained by running Alg. 1. for a certain number of iterations. However, I still think that the paper presents a solid contribution and I support the paper for acceptance. Unfortunately, there are some negative reviews; perhaps, it would be helpful if these reviewers could kindly let us know whether their concerns were resolved or not.

---

> > > ### Author Response · Authors · 2021-11-28
> > > **Thank You for the Response!**
> > >
> > > Thank you for your appreciation of our work. For your question about certain number of iterations in Alg. 1, we will supplement detailed experiments in our final edition.

---

### Official Review · Reviewer_AGhv · 2021-11-01

**Correctness:** 3
**Technical Novelty And Significance:** 2
**Empirical Novelty And Significance:** 2
**Recommendation:** 3
**Confidence:** 4

**Main Review:**

strengths:
1.  Result in theorem 1 seems useful as the bound depends on m.
2. Result in theorem 2 is indeed helpful in motivating the intuitive landmark selection strategy.
3. Improvements in fig2 seem encouraging.

Critical comments:
1. The paper seems to present a parallel solver rather than a distributed one. This is because a run of the solver with entire data is also a sub-step. Simulations are also performed with multiple cores rather than nodes. I think this must be highlighted in title and abstract too. Currently, it seems that the paper is presenting a distributed algorithm, which is not the case.

2. The tree structured based parallel strategy has limited novelty as such ideas are popular among parallel methods.

3. Some important details are missing in Algo1 (and elsewhere). For e.g., it is mentioned that solutions are concatenated to initialize the next level solvers. What exactly is meant by concatenation ? is it averaging ? There are works that study various aggregation strategies (e.g., https://ieeexplore.ieee.org/stamp/stamp.jsp?arnumber=8911318 etc.). Also step14 in algo1 must be return \alpha^1.

4. Speed up is achieved primarily because for solver instances with higher data get better initialization. Since this does not improve the rate of convergence and only may effect the involved constants, the method is intended to achieve limited improvement. It would be insightful if improvement of the constants involved in rate of convergence are theoretically bounded etc.

5. Since the solver runs with entire data too, and the problem is convex, convergence is anyway guaranteed.

6. Since partition strategy is one of the contributions, it would have been nice to compare empirically the proposed strategy with baselines in terms of computation-accuracy trade-off.

7. Section 3.3 is repetition of Lee et.al, 17 and hence can go into background/related work.

8. What are the baselines Ca-ODM, DiP-ODM and DC-ODM? References seem to be missing. Without such details it is hard to evaluate the simulations section.

9. Since acc-DiODM is Lee et.al.'17, the section 4.2 may be postponed to appendix.

10. It is not clear what  the baselines in figure 3 are. Again this makes evaluation of simulations section difficult.

11. A major weakness of simulation section is comparison with state-of-the-art distributed-SVM solvers is missing. This seems very important as the form of the optimization is same as that in SVM. For example, https://arxiv.org/pdf/1005.2012.pdf etc.

**Summary Of The Paper:**

The paper presents a parallel solver for speeding up training time of ODM. The key idea is to perform computation via a tree structure. Solvers at parent level are initialized with "concatenated" (averaged?) solutions from child level. The hope is that convergence will be faster because of closeness to optimal though the data size grows.

 A Schur complement based strategy for partitioning for the data is presented.

For linear kernels, the DSVRG solver of Lee et.al.'17 is (directly) employed. Simulations show that for linear kernel, the Lee et.al., outperforms the proposed method.

However, with non-linear kernels, the proposed methods improves over some baselines.

**Summary Of The Review:**

Overall, the methodology seems to have limited novelty and limited potential for improvement over baselines. Important details seems to be missing in the presentation. Simulations seem to be weak.

---

> ### Author Response · Authors · 2021-11-15
> **Response to reviewer AGhv**
>
> We thank the reviewer for the comments. We have revised the paper according to the suggestions and would like to clarify several things:
>
> [Q] The paper seems to present a parallel solver rather than a distributed one. This is because a run of the solver with entire data is also a sub-step. Simulations are also performed with multiple cores rather than nodes. I think this must be highlighted in title and abstract too. Currently, it seems that the paper is presenting a distributed algorithm, which is not the case.
>
> [A] In reply to “Simulations are also performed with multiple cores rather than nodes”, our experiments is performed on a Spark cluster which claimed in our experiment settings. The cores number mentioned actually means the whole cores used in our cluster.
>
> It is noteworthy that our algorithm has an early stop mechanism described in line 5-7 of Alg. 1 to avoid a run of the solver with entire data. In our algorithms, this mechanism is performed when the change of the solution is less than a given threshold during the dual coordinate descent process after partition combined.
>
> [Q] Some important details are missing in Algo1 (and elsewhere). For e.g., it is mentioned that solutions are concatenated to initialize the next level solvers. What exactly is meant by concatenation ? is it averaging ? … Also step14 in algo1 must be return \alpha^1.
>
> [A] We aim to solve ODM in dual form in Alg.1. In dual form of ODM, each instance is corresponding to a solution. Therefore, the word “concatenate” means concatenate two vectors.
>
> Strategy given by reviewer is designed to accelerate the process of first-order gradient optimization algorithm, which cannot be implemented to solve ODM in dual form.
>
> $\alpha^l$ in step 14 is correct since our early stop mechanism mentioned above.
>
> [Q] Speed up is achieved primarily because for solver instances with higher data get better initialization. Since this does not improve the rate of convergence and only may effects the involved constants, the method is intended to achieve limited improvement. It would be insightful if improvement of the constants involved in rate of convergence are theoretically bounded etc.
>
> [A] Distributed optimization mainly focus on how to minimize a global function in parallel. Since distributed methods cannot outperform their local versions on convergence rate, this problem is not concerned by distributed optimization. The convergence rate of convex quadratic programming has been studied in many previous works (Boyd & Vandenberghe, 2014).
>
> [Q] Since partition strategy is one of the contributions, it would have been nice to compare empirically the proposed strategy with baselines in terms of computation-accuracy trade-off. What are the baselines Ca-ODM, DiP-ODM and DC-ODM? References seem to be missing. Without such details it is hard to evaluate the simulations section.
>
> [A] We survey these methods in introduction. We have supplied these citations and feel sorry to miss the citations of Ca-ODM, DiP-ODM and DC-ODM in our experiment part. These methods are state-of-the-art distributed solver designed for SVM. We compare these methods to show the superiority of our proposed strategy.
>
> [Q] It is not clear what the baselines in figure 3 are. Again this makes evaluation of simulations section difficult.
>
> [A] We survey these methods in introduction. We have supplied these citations and feel sorry to miss the citations of SVRG and CSVRG in our experiment part. SVRG and CSVRG are two off-the-shelf methods for solving ODM in local environment. We evaluate these methods to show the speedup in distributed environment.
>
> [Q] A major weakness of simulation section is comparison with state-of-the-art distributed-SVM solvers is missing. This seems very important as the form of the optimization is same as that in SVM. For example, https://arxiv.org/pdf/1005.2012.pdf etc.
>
> [A] We perform the comparison between our method and other state-of-the-art distributed solver designed for SVM in Section 4.2.
>
> [1] S. P. Boyd and L. Vandenberghe. Convex Optimization. Cambridge, United Kingdom: Cambridge University Press, 2014.

---

### Official Review · Reviewer_cre7 · 2021-11-03

**Correctness:** 4
**Technical Novelty And Significance:** 2
**Empirical Novelty And Significance:** 2
**Recommendation:** 3
**Confidence:** 3

**Main Review:**

One of my concerns is that the optimization problem for ODM is quite close to the standard L1/L2-SVMs. As the authors mentioned, there are a lot of previous results on distributed solvers for SVMs. It seems to me that this paper proposes a distributed solving method for a similar problem to SVM(ODM). The authors should clarify the differences between the proposed method and previous results on distributed SVMs, so that the technical contribution is clear.

**Summary Of The Paper:**

The paper considers the optimization problem for ODM and proposes a distributed method for solving the problem. The authors prove its convergence to the optimum and show some experimental results comparing the proposed methods and previous work.

**Summary Of The Review:**

The problem seems quite similar to the standard SVMs and a new distributed solver for the problem looks not significantly new.

---

> ### Author Response · Authors · 2021-11-15
> **Response to Reviewer cre7**
>
> We thank the reviewer for the comments. We have revised the paper according to the suggestions and would like to clarify several things:
>
> [Q] The optimization problem for ODM is quite close to the standard L1/L2-SVMs… The authors should clarify the differences between the proposed method and previous results on distributed SVMs, so that the technical contribution is clear.
>
> [A] We have clearly clarified the differences in our paper:
>
> “Although many distributed solver for quadratic programming (QP) problems can be directly ap- plied to ODM, these off-the-shelf solvers all ignore the intrinsic structure of the problem and can hardly achieve the greatest efficiency.” “Up to now, most partition strategies utilize the clustering algorithms to form the partitions… However, ODM heavily depends on the mean and variance of training data. Directly treating clusters as partitions will lead to significant difference among the distribution of partitions and the whole data, which makes the local solutions on each partition are far from the global one.”
>
> Based on above analysis, we also present an upper bound in theorem 2 to guarantee the efficiency of our proposed method.
>
> In distributed environment, it is of great significance to design specific distributed solver since general solver can hardly get the optimal solution in all convex quadratic problems. To name a few, Yin et al. (2021) and Yin et al. (2020)are specifically designed for kernel ridge regression, Kim et al. (2020) and Nguyen et al. (2016) are specifically designed for logistics regression.
>
> [1] R. Yin, Y. Liu, W. Wang and D. Meng. Distributed Nyström Kernel Learning with Communications. In Proceedings of the 38th International Conference on Machine Learning, pp.12019-12028, Virtual Event, 2021.
>
> [2] R. Yin, Y. Liu, L. Lu, W. Wang and D. Meng. Divide-and-Conquer Learning with Nyström: Optimal Rate and Algorithm. In Proceedings of the 35th AAAI Conference on Artificial Intelligence, pp. 6696-6703, New York, NY, 2020.
>
> [3] M. Kim, J. Lee and L. Ohno-Machado and X. Jiang. Secure and Differentially Private Logistic Regression for Horizontally Distributed Data, IEEE Transactions on Information Forensics and Security, vol. 15, pp. 695-710, 2020.
>
> [4] V. Nguyen, T. D. Nguyen, T. Le, S. Venkatesh and D. Q. Phun. One-Pass Logistic Regression for Label-Drift and Large-Scale Classification on Distributed Systems, In Proceedings of the 16th IEEE International Conference on Data Mining, pp. 1113-1118, Barcelona, Spain, 2016.

---

### Decision · Program_Chairs · 2022-01-20

**Decision:**

Reject

**Comment:**

While some of the reviewers find that the paper proposes a solid contribution to a problem, I will tend
to agree with other ones that the proposed approach has limited novelty and limited potential for improvement over baselines. In addition,  simulations are pretty weak due to lack of comparisons to strong baselines and to lack of clarity.